# Structural assembly of the megadalton-sized receptor for intestinal vitamin B$_{12}$ uptake and kidney protein reabsorption

Casper Larsen[1], Anders Etzerodt [1], Mette Madsen[1], Karsten Skjødt[2], Søren Kragh Moestrup[1,2,3] & Christian Brix Folsted Andersen[1]

The endocytic receptor cubam formed by the 460-kDa protein cubilin and the 45-kDa transmembrane protein amnionless (AMN), is essential for intestinal vitamin B$_{12}$ (B$_{12}$) uptake and for protein (e.g. albumin) reabsorption from the kidney filtrate. Loss of function of any of the two components ultimately leads to serious B$_{12}$ deficiency and urinary protein loss in humans (Imerslund-Gräsbeck's syndrome, IGS). Here, we present the crystal structure of AMN in complex with the amino-terminal region of cubilin, revealing a sophisticated assembly of three cubilin subunits combining into a single intertwined β-helix domain that docks to a corresponding three-faced β-helix domain in AMN. This β-helix-β-helix association thereby anchors three ligand-binding cubilin subunits to the transmembrane AMN. Electron microscopy of full-length cubam reveals a 700–800 Å long tree-like structure with the potential of dimerization into an even larger complex. Furthermore, effects of known human mutations causing IGS are explained by the structural information.

[1] Department of Biomedicine, Aarhus University, 8000 Aarhus C, Denmark. [2] Department of Cancer and Inflammation, Institute of Molecular Medicine, University of Southern Denmark, 5000 Odense C, Denmark. [3] Department of Clinical Biochemistry and Pharmacology, Odense University Hospital, 5000 Odense C, Denmark. Correspondence and requests for materials should be addressed to C.B.F.A. (email: cbfa@biomed.au.dk)

Mammals require $B_{12}$ (or $B_{12}$-derived analogues) as coenzyme in two essential enzymatic reactions catalysed by methionine synthase and methylmalonyl-CoA mutase. As biosynthesis of $B_{12}$ is confined to prokaryotes, mammals depend on dietary supply of the vitamin. Consequently, a sophisticated $B_{12}$ uptake and transport mechanism has evolved[1]. The intestinal uptake of $B_{12}$ is a complex process that requires binding to the carrier protein 'intrinsic factor' (IF) and uptake of IF-$B_{12}$ by the intestinal cubam receptor[2,3]. Defects in components involved in $B_{12}$ uptake, such as acquired deficiency of IF that is a common disease in the elderly population[4] and loss-of-function mutations of cubam (IGS)[5] leads to serious disease characterised by megaloblastic anaemia and neurological symptoms. Besides its role in intestinal IF-$B_{12}$ uptake, cubam is a major component of the kidney proximal tubule epithelium, where it is responsible for reabsorption of abundant proteins in the renal ultrafiltrate, such as apolipoprotein A-1[6,7], transferrin[8], albumin[9] and haemoglobin[10]. Hence, defects in cubam are also associated with urinary loss of proteins.

Cubilin, the ligand-binding component of cubam, consists of an N-terminal region (residues 36–131) encompassing a coiled-coil region followed by eight epidermal growth factor-like (EGF-like) domains and 27 CUB (for complement C1r/C1s, Uegf, Bmp1) domains[3]. Cubilin has been suggested to form trimers[11,12], which are anchored to the apical membrane via interaction with the type-1 transmembrane protein amnionless (AMN)[13]. AMN is formed by an extracellular domain, a transmembrane helix and a short cytoplasmic domain harbouring two adaptor protein-2-binding signals (Phe-X-Asn-Pro-X-Phe) for ligand-independent internalisation in clathrin-coated pits[13,14]. As cubilin and AMN depend on each other for proper processing and translocation to the apical membrane, disrupted interaction of the two proteins leads to retention of both in the endoplasmic reticulum (ER)[13,15,16]. Accordingly, defects in the genes encoding either cubilin or AMN both give rise to IGS.

To uncover the structural basis for AMN-mediated anchoring of cubilin to the apical membrane and mutations that cause IGS, we have determined the crystal structure of the ectodomain of AMN in complex with the N-terminal part of cubilin. The structure reveals a receptor architecture, where three cubilin chains combine into an intertwined β-helical structure that docks on to a corresponding β-helix domain in AMN. The AMN mutations that cause IGS are located in critical areas of the structure and either disrupt the cubilin–AMN interface or destabilise the structure of AMN. An exception is the T41I mutation, which does not appear to have a direct effect on the structure of AMN or its association with cubilin. Here, we show that the T41I mutation alters the glycosylation pattern of AMN, which reduces the expression levels of cubam on the cell surface.

## Results

**Structure determination**. For structural analysis, the ectodomain of AMN (20–357) and an N-terminal fragment of cubilin (26–135) were co-expressed in *Escherichia coli*. Crystals of the AMN(20–357)–cubilin(26–135) complex were obtained and the structure determined to 2.3 Å resolution by the single anomalous diffraction method (Table 1). Electron density is observed for the entire ectodomain of AMN and for cubilin residues 37–120. The final model displays good crystallographic and geometric statistics with $R_{work}/R_{free} = 0.21/0.23$.

**Structure of AMN(20–357)**. AMN forms a bean-shaped structure protruding ~60 Å from the plasma membrane and is composed of four structural entities (Fig. 1). A SEA (for Sperm protein, Enterokinase and Agrin) domain is located in the C-

**Table 1 Data collection, phasing and refinement statistics**

|  | Native | SeMet |
|---|---|---|
| **Data collection** |  |  |
| Space group | I222 | I222 |
| Cell dimensions |  |  |
| *a, b, c* (Å) | 70.3, 158.8, 237.2 | 70.3, 159.1, 237.6 |
| *α, β, γ* (°) | 90, 90, 90 | 90, 90, 90 |
| Wavelength | 0.976 | 0.980 |
| Resolution (Å) | 50–2.3 (2.4–2.3) | 50–2.5 (2.6–2.5) |
| $R_{sym}$ | 0.089 (1.652) | 0.103 (1.474) |
| $I/\sigma I$ | 19.53 (2.10) | 11.89 (1.12) |
| Completeness (%) | 100.0 (100.0) | 100.0 (100.0) |
| Redundancy | 13.31 (13.85) | 6.98 (7.04) |
|  |  |  |
| **Refinement** |  |  |
| Resolution (Å) | 2.3 |  |
| No. of reflections | 59,436 |  |
| $R_{work}/R_{free}$ | 0.212/0.230 |  |
| No. of atoms |  |  |
| Protein | 4453 |  |
| Water | 102 |  |
| *B*-factors |  |  |
| Protein | 84.03 |  |
| Water | 64.79 |  |
| R.m.s. deviations |  |  |
| Bond lengths (Å) | 0.004 |  |
| Bond angles (°) | 0.795 |  |

Values in parentheses are for highest-resolution shell

terminal part of AMN directly preceding the transmembrane helix[17]. The SEA domain has a classic βαββαβ-fold[18] found in various types of proteins including initiation and elongation factors[19]. A cysteine-rich region links the SEA domain to the N-terminal part of AMN. With the exception of a single α-helix the cysteine-rich region has no apparent secondary structure. The fold of the two N-terminal domains of AMN (denoted β-helix 1 and β-helix 2) is similar and is formed by right-handed β-helices with hydrophobic cores. Both domains are shaped as inverted three-faced pyramids but they differ substantially in the base regions. The base of β-helix 1 is engaged in interactions with cubilin, whereas the base of β-helix 2 is capped by two amphi-pathic α-helices that segregate the hydrophobic core of the domain from the solvent.

AMN has ~3 kDa of Asn-linked oligosaccharides as demonstrated by PNGase F digestion of AMN purified from solubilised human kidney membranes[13]. Two consensus sequences (Asn-X-Ser/Thr) are present in AMN for potential attachment of N-linked oligosaccharides: Site I for glycosylation of Asn35 and site II for glycosylation of Asn39. Both Asn35 and Asn39 are located in the apex of β-helix 1 (Fig. 1d). Asn35 is buried within β-helix 1 and engages in hydrogen bonding with Asn27 and Arg109. As Asn35 is not solvent exposed, attachment of an oligosaccharide would require major rearrangements of AMN β-helix 1. Asn39 on the other hand is solvent exposed and points towards the SEA domain. A mature N-linked glycosylation can be modelled on Asn39 without any clashes or rearrangements (Fig. 1e).

**Structure of cubilin**. Three cubilin (residues 26–135) chains join to form a pin-shaped molecule with an approximate diameter of 30 Å and a length of 80 Å (Fig. 1a). The head is formed by a three-turn intertwined β-helix, which irreversibly interlocks three cubilin chains. The β-helix is extended with a single β-hairpin at the N-terminal of each chain. In the C-terminal region three cubilin chains form a coiled-coil, which is interrupted midway owing to the presence of a proline residue (Pro103). This

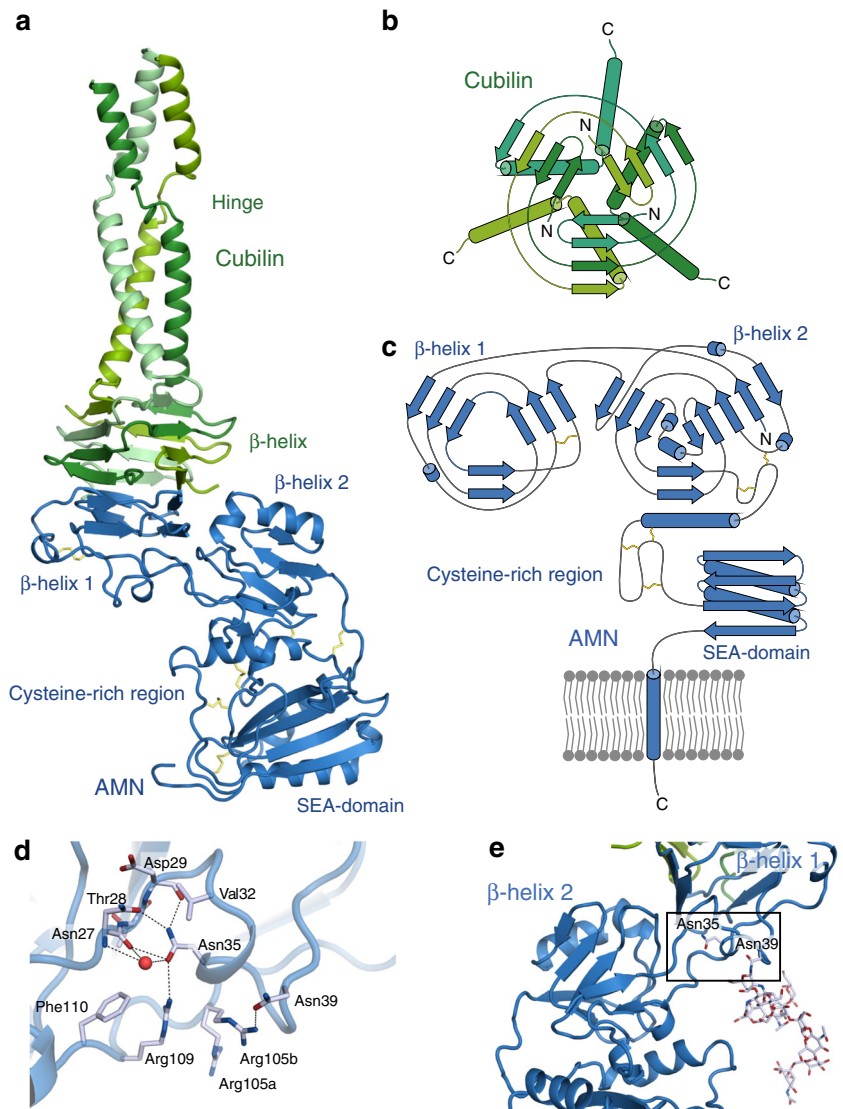

**Fig. 1** Structure of AMN-Cubilin. **a** Cartoon representation of the crystal structure of ectoAMN (residues 20–357, blue) in complex with trimeric cubilin (residues 36–135, shades of green). Disulphide bonds are shown as yellow sticks. **b** Topology diagram of the cubilin trimer. **c** Topology diagram of AMN. **d** Close-up views of selected residue surrounding the potentially N-linked glycosylated AMN residues, Asn35 and Asn39. Hydrogen bonds and ionic interactions are marked by dashed black lines. Water molecules are shown as red spheres. **e** N-linked glycosylation modelled on Asn39. The black box represents the close-up view shown in **d**

hinge-region introduces a certain degree of flexibility in the coiled-coil region as we observe a dramatic increase of B-factors in the region following the hinge (Supplementary Figure 1). A potential glycosylation site on Asn105 is also located in the hinge region and oligosaccharides can be modelled on Asn105 with minor rearrangements (Supplementary Figure 2). No interpretable electron density is observed for cubilin residues 121–135, which is predicted to extend the coiled-coil region leading up to the first EGF-like domain. These residues will extend the length of the cubilin N-terminal region by 15 Å giving it a total length of 75 Å.

**AMN–cubilin interface.** The cubilin interface with AMN is formed by the N-terminal strands of three cubilin chains (residues 42–49), which combine into a triangular face with a hydrophobic centre formed by Met44 (Fig. 2a). This structure is complementary to the triangular base of AMN β-helix 1. Here the

hydrophobic centre is formed by AMN residues Met69, Leu71, Leu77, Leu79 and Phe85 (Fig. 3b). The two hydrophobic faces form extensive Van der Waals interactions shielding the hydrophobic areas from the solvent. In addition, the β-strands at the triangle perimeter form anti-parallel β-sheets via main-chain hydrogen bonds (Fig. 2c–e). Only few interactions are formed between the side chains of cubilin and AMN. Although the structure of the three cubilin chains that constitute the AMN interface are more or less identical and display three-fold symmetry, their interactions with AMN are markedly different (Fig. 2c–e). This is most likely caused by the asymmetric nature of the AMN β-helix 1. Alignment of AMN and cubilin sequences from distantly related organisms reveals that the hydrophobic residues forming the cores of the β-helical domains of both AMN and cubilin are highly conserved (Fig. 3a–d). This suggests that the cubam receptor architecture with three cubilin chains anchored to AMN via β-helix–β-helix association is conserved among the species.

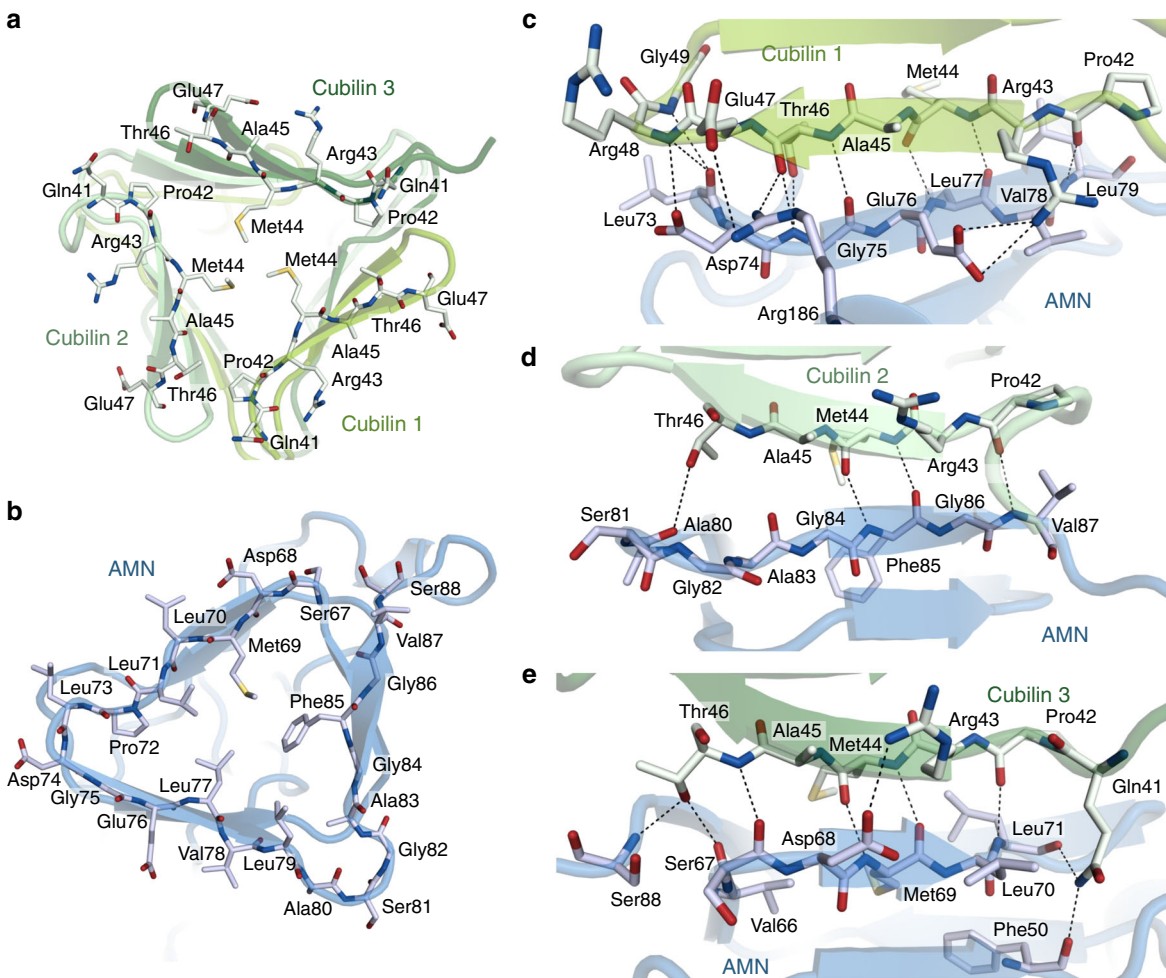

**Fig. 2** Interaction between AMN and cubilin. **a** AMN interface of the trimeric cubilin β-helix. The hydrophobic centre of the β-helix is formed by Met44 residues. **b** Cubilin interface of AMN. AMN in **b** is related to cubilin in **a** by a vertical rotation of 180°. **c–e** Interactions between the three individual polypeptide chains of cubilin with AMN. Residues directly involved in interactions are shown as sticks. Hydrogen bonds and ionic interactions are marked by dashed black lines

**Electron microscopy (EM) on full-length cubam**. In order to visualise the entire cubam receptor, we purified full-length cubam from solubilized porcine kidney membranes using immobilised human IF-B$_{12}$ and performed negative-stain EM. In the presence of Ca$^{2+}$, the electron micrographs reveal 700–800 Å long tree-shaped structures with an ~400 Å long stem and a globular crown-region (Fig. 4a, c). Two distinct populations of receptors are observed (marked by arrows in Fig. 4a), a population with a single stem and a small crown and a population with two more or less parallel stems and a larger crown, suggesting dimerisation of the receptor complex. When ethylenediaminetetraacetic acid (EDTA) is added, the electron micrographs show a markedly different morphology (Fig. 4b). These micrographs resemble previously published electron micrographs of cubilin[11] (probably in complex with AMN) and show partially unfolded structures with the three cubilin chains as individual lobes connected via interaction with AMN. These results show that Ca$^{2+}$-binding is important for the structural integrity of the receptor by stabilising individual domains (CUB and EGF-like) and/or by mediating interactions between cubilin chains.

The observed stem of cubam is likely formed by AMN and the cubilin N-terminal region including the eight EGF-like domains. EGF-like domains are small compact domains composed of around 40 residues and are often present as consecutive repeats[20]. A consensus Ca$^{2+}$-binding motif is frequently found in the

interface between adjacent EGF-like domains and may serve to stabilise the conformation of two consecutive domains[21]. In cubilin, consensus Ca$^{2+}$-binding motifs are present in the interfaces between EGF-like domains 1–2, 3–4, 4–5 and 7–8. Several structures have shown that consecutive EGF-like domains form rod-like structures with almost linear arrangement[22–25]. Based on the structure of human Notch-1 we have modelled the eight EGF-like domains of cubilin and attached them to the structure of AMN–cubilin (Fig. 4d). The resulting model has a length of ~400 Å, which perfectly matches the length of the stem observed in the electron micrographs. The physiological role of the stem might well be to place the ligand-binding regions further from the membrane, which may be advantageous for catching multiple ligands in the fluid passing the apical cells.

The crown of cubam in the electron micrographs is most likely constituted by the 27 CUB domains from each cubilin subunit. From the micrographs it appears that the CUB domains are organised in a more or less ordered structure that likely exposes the ligand-binding domains to the surrounding fluids, while other domains are packed in the interior.

**IGS mutations**. So far, 69 IGS causing mutations have been described (Human Gene Mutation Database version 2017.1[26]). Non-sense mutations that introduce pre-mature stop codons or

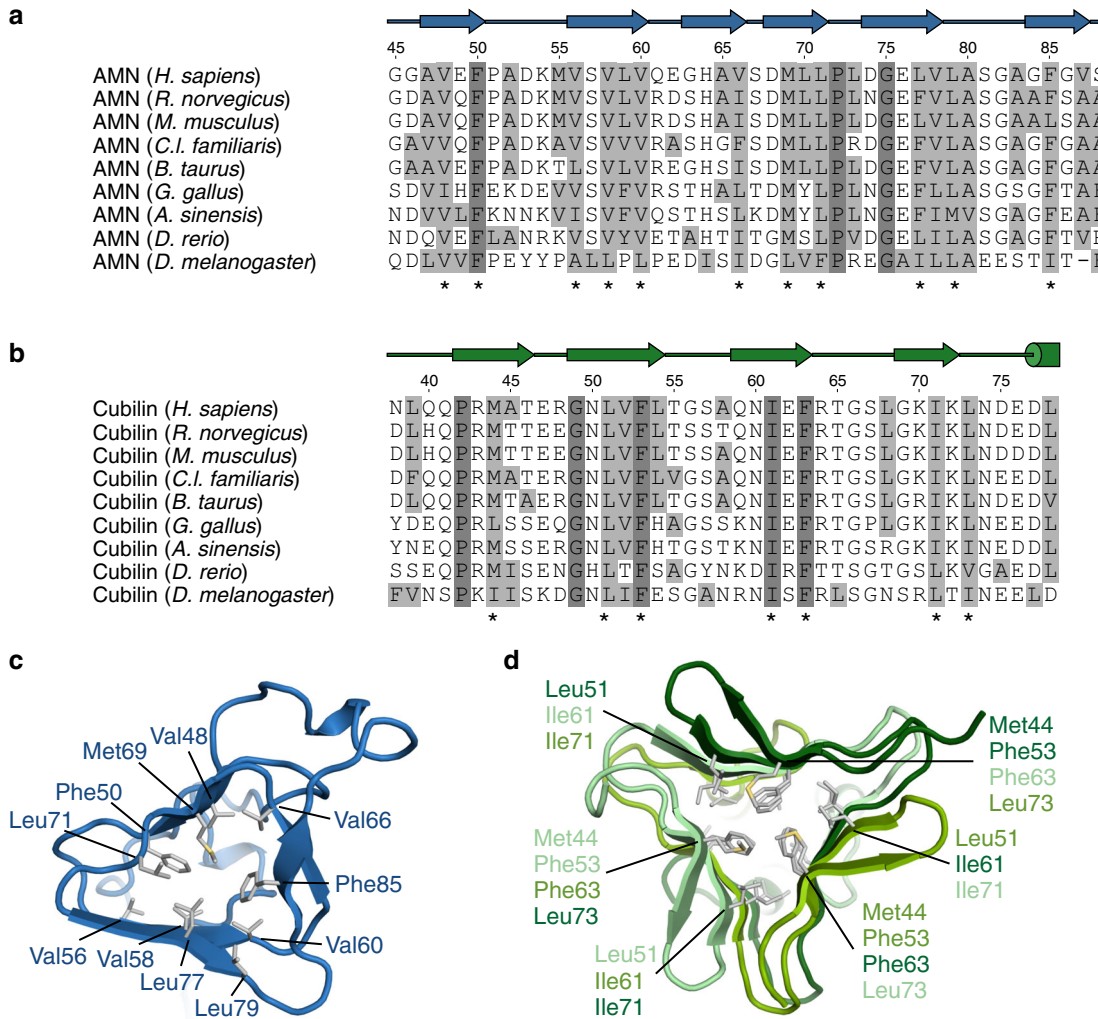

**Fig. 3** Sequence alignment of AMN and cubilin β-helices. **a** Sequence alignment of AMN β-helix 1. **b** Sequence alignment of the cubilin β-helix. Secondary structure elements are shown above the alignment. Fully conserved residues are marked by dark grey boxes. Hydrophobic residues (Ala, Val, Leu, Ile, Met, Phe) are marked by light grey boxes. Residues constituting the core of the β-helices are marked by asterisks below the alignment. **c** and **d** Cartoon representation of AMN β-helix 1 (**c**) and the cubilin β-helix (**d**). Hydrophobic residues forming the core of the β-helices are shown as grey sticks

mutations that affect splicing-events readily explain absence of cubam expression[27]. However, except for those mutations in cubilin that directly affects the IF-B$_{12}$-binding region[12,28], less is known about the causes of missense mutations leading only to a single amino acid substitution (Supplementary Table 2).

A single amino acid substitution that causes IGS is the AMN T41I missense mutation[29]. Expression of the AMN T41I mutant and cubilin in *E. coli* yields a stable complex (Supplementary Figure 5). Hence, receptor malfunction caused by T41I is not clearly explained by the structural changes alone. As described above, AMN contains two consensus sequences for potential N-linked glycosylation. The T41I mutation alters site II and consequently inhibits the transfer of oligosaccharides by oligo-saccharyltransferases in the ER to Asn39[30] (Fig. 5a). In order to investigate the functional significance of the two potential glycosylation sites, we performed flowcytometry and immuno-precipitation experiments on various AMN site I and site II mutants co-transfected with cubilin in CHO cells (Fig. 5b, c).

AMN N35Q and S37A mutations both disrupt N-glycosylation site I, however, their effect on cubam surface expression is markedly different. Whereas N35Q completely abolishes surface expression, the AMN S37A mutant behaves as wild type. This

indicates that Asn35 is not glycosylated, but instead important for intra-molecular interactions as also suggested from its position in the structure of AMN (Fig. 1d). The AMN T41I mutation that disrupts site II causes a significant reduction of surface expression (Fig. 5b) explaining why this mutation causes IGS. Interestingly, when inhibiting both N-glycosylation site I and II using the S37A/T41I double mutant, surface expression is restored (Fig. 5b). This suggest an interplay between the two adjacent glycosylation sites, which is supported by migration of AMN in SDS–PAGE (Fig. 5c). Here, neither of the individual S37A and T41I mutations causes a reduction in the apparent molecular size, whereas only the S37A/T41I double mutation migrates similar to PNGaseF treated AMN. Altogether, this indicates that the T41I mutation causes an aberrant glycosylation pattern of AMN ultimately leading to reduced surface expression of cubam.

The AMN IGS mutations L59P[27], M69K[31], C234F[32] and G254E[27] are all retained in the ER when co-expressed with cubilin in HEK293 cells[16], which explains why these mutations impair cubam receptor function and cause IGS. Introducing the individual mutations in our *E. coli* expression system does not yield any soluble AMN or AMN–cubilin complex (Supplementary Figure 5). Since the folding of AMN and interaction with

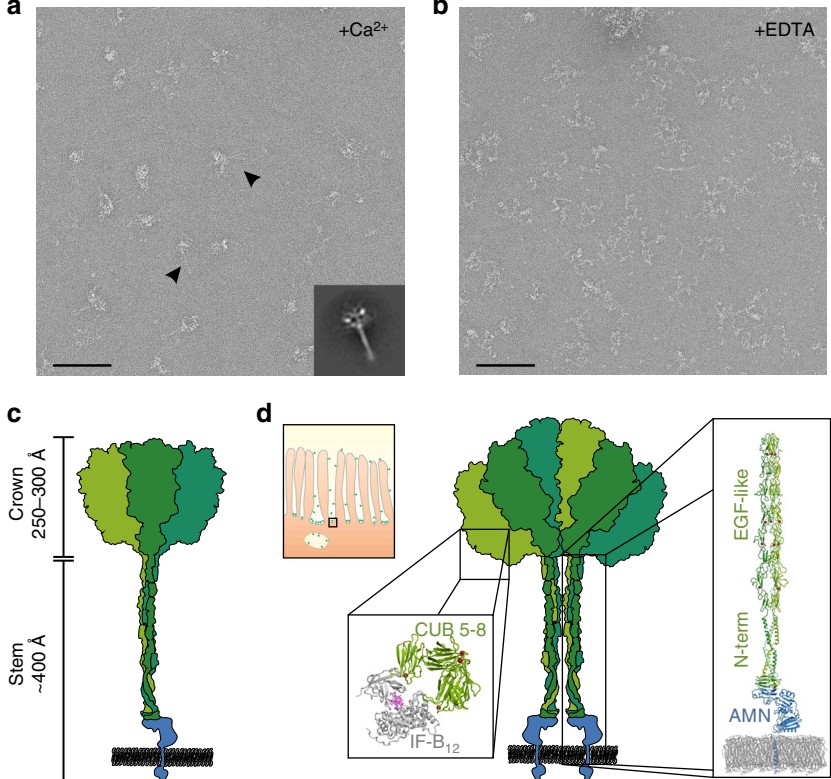

**Fig. 4** Negative stain electron microscopy and modelling of full-length cubam. **a** Typical electron micrograph showing two different forms of cubam (marked by arrows)—a single-stem form with a small head domain and a double-stem form with a larger head domain. A 100 nm scale bar is shown in the lower left-hand corner. Class average of 758 particles of the double-stem form is shown in the lower right-hand corner. Meaningful class-averages of the single-stem form of cubam could not be obtained due to inhomogeneous particles. **b** Typical electron micrograph showing partly unfolded cubam in the presence of 10 mM EDTA. $Ca^{2+}$-binding motifs are found in several of the CUB and EGF-like domains of cubilin. Receptor unfolding in the presence of EDTA suggests that $Ca^{2+}$-binding serves to maintain the structural integrity of cubam. **c** Schematic representation of the single-stem form of cubam with approximate dimensions of "stem" and "crown" regions. **d** Schematic representation of the double-stem form of cubam. The top-left box shows a schematic representation of the cubam location the enterocyte brush-border membrane with approximate proportional sizes of cubam and the microvilli. The predicted IF-$B_{12}$-binding region of cubam is marked by a box and the structure of CUB$_{5-8}$-IF-$B_{12}$[12] is shown as enlargement with the $B_{12}$ structure visualised in the original purple colour. We anticipate that the stem of cubam is formed by AMN and the C-terminal region of cubilin including the eight EGF-like domains. The stem of cubam is marked by a box and a model of the stem is shown as enlargement

cubilin are mutually dependent on each other, we cannot decipher the cause of receptor malfunction from these experiments. However, we can predict the consequences of the individual mutations from the structure of AMN.

The AMN mutations L59P and M69K, are both located in the β-helix 1 domain responsible for the interaction with cubilin (Fig. 6a). Leu59 is located in the β-helical core of the domain and mutation to proline alters the main chain conformation of the residue and introduces clashes with Ser149 (Fig. 6b). This probably leads to a general destabilization of the entire domain and disruption of the AMN–cubilin interface. Met69 is positioned directly in the AMN–cubilin interface where it engages in hydrophobic interactions (Fig. 6b). Mutation to a lysine residue in this position introduces an unfavourable positive charge in the hydrophobic interface that most likely prevents the AMN–cubilin association.

The two AMN missense mutations C234F and G254E are located in regions further from the cubilin-binding site (Fig. 6a). The C234F mutation disrupts a disulphide bond in the cysteine-rich region and leads to a free cysteine residue, which can engage in unspecific disulphide bonds and cause misfolding of AMN. The G254E mutation is located in the SEA domain. Substitution of Gly254 with a glutamate residue will introduce clashes and position the charged glutamate residue in a hydrophobic

environment (Fig. 6d) that will probably destabilise the entire domain and result in misfolded AMN.

## Discussion

The structural analysis of cubam reveals an unexpected receptor architecture based on β-helix–β-helix association of AMN and three subunits of cubilin. β-helices are best characterised in prokaryotic organisms, where they mediate various functions, including structural organisation of the cell interior[33,34]. The β-helices are well-known to form end-to-end associations and unless capped by amphipathic α-helices as observed in β-helix 2 of AMN, the open-ends of β-helices tend to cause aggregation or unspecific interactions with β-sheet structures from other proteins[35]. This phenomenon could explain the mutual dependence of cubilin and AMN for processing and translocation to the cell surface. Thus, if the interaction partner is not available, the open-ended β-helices may cause nonspecific interactions that abrogate folding and, as a consequence, the proteins are retained in the ER.

Running a DALI search with the structures of the β-helical domains of AMN and cubilin as templates did not disclose any homologues structures. However, browsing through β-helical structures deposited in the RCSB PDB (www.rcsb.org)[36], we discovered an unexpected homology to the tailspike proteins

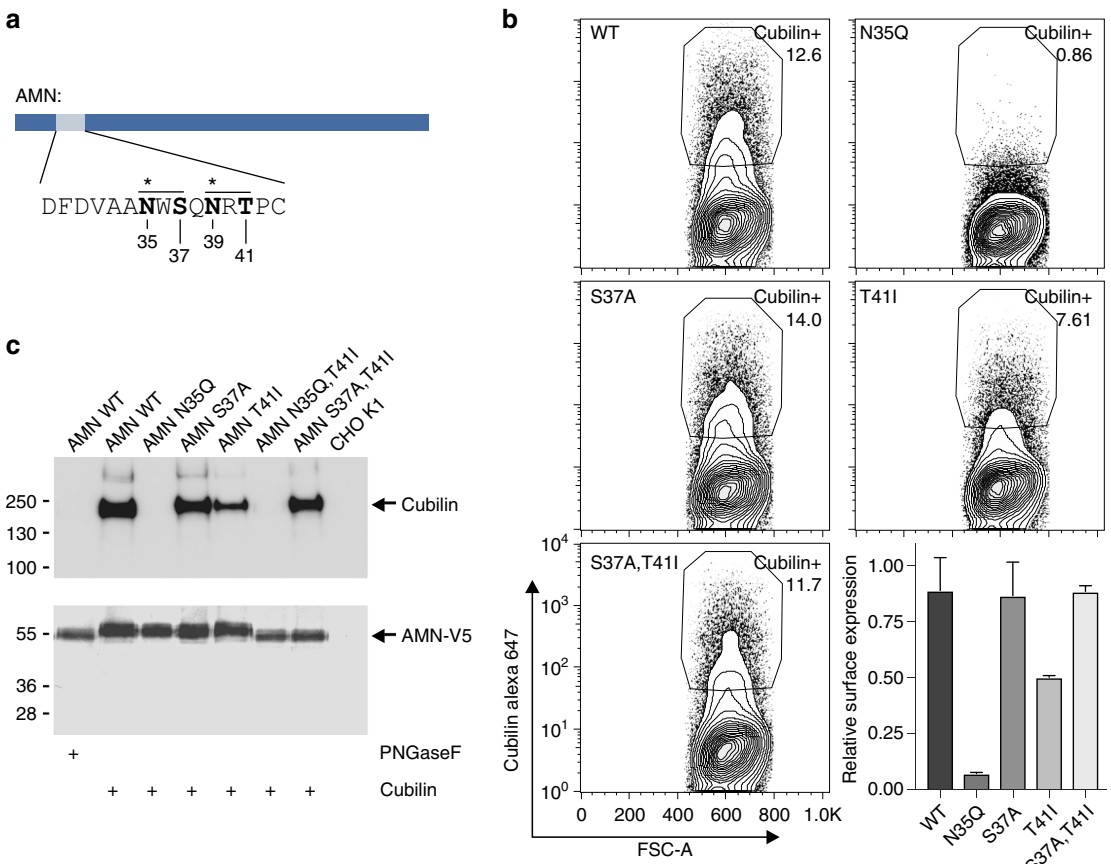

**Fig. 5** N-linked glycosylation of AMN. **a** Consensus sequence motif for potential N-linked glycosylation of AMN. The two consensus sites are marked by lines above the sequence. Potentially glycosylated Asn residues are marked by asterisks. Residues selected for site-directed mutagenesis are marked in bold. **b** Surface expression of cubilin in transiently transfected CHO K1 cells. Cells were co-transfected with cubilin and AMN wild type or AMN mutants. Live cells were gated in SSC-A and FSC-A and replotted in a contour plot showing Cubilin expression vs. FSC-A. Correct gate for Cubilin expressing cells (Cubilin+) was set based on non-cubilin expressing control cells (mock transfected). Full gating strategy is shown in Supplementary Figure 3A. Relative surface expression of cubilin is shown in the bottom right corner. As cubilin and AMN expression was established by transient transfection only 10–20% of analysed cells co-expressed both proteins (Supplementary Figure 3B). The relative surface expression of cubilin was therefore calculated as % of cubilin+ cells (surface expression)/% cubilin+, AMN+ cells (total cell stain), experiment was performed in triplicates and error bars represent standard error of mean. **c** Immunoprecipitation of cubilin with wild type or mutant forms of AMN-V5. The top blot was visualized using rabbit polyclonal anti-rat cubilin antibody followed by Horse-radish peroxidase conjugated goat polyclonal anti-rabbit IgG. The bottom blot was visualized using mouse monoclonal anti-V5 alkaline phosphatase (AP) conjugated antibody. Uncropped blots are shown in Supplementary Figure 4

from bacteriophages[37] and the functionally analogues bacterial type VI secretion system[38]. In particular, the spike proteins from the *Vibrio cholerae* secretion system[39] is remarkably similar to the structure of AMN β-helix 1 and the β-helix of cubilin. In both structures, an intertwined three-chained β-helix associate end-to-end with a pyramid-shaped β-helix (Fig. 7). However, in contrast to the cubilin β-helix, the *V. cholerae* spike has a channel in its centre for translocation of toxic effector molecules into the penetrated cell.

As AMN and cubilin are mutually dependent on each other for translocation of a functional cubam receptor to the apical surface of the cell, mutations disturbing folding of either protein or preventing their association cause impaired $B_{12}$ uptake and ultimately IGS. With the present structural information, we estimate the effects caused by known IGS missense mutations in AMN. The AMN IGS mutations L59P, M69K, C234F and G254Q are all located in critical regions of AMN explaining their impact on cubam processing and translocation. The T41I mutation on the other hand is located in a loop region and does not appear to have a direct effect on folding of AMN or its interaction with cubilin.

Instead, T41I disrupts the consensus site for glycosylation of Asn39. N-linked glycosylations play an important role in the folding of proteins in the ER, where the attachment of oligosaccharides not only improves the kinetics and thermodynamics of protein folding, but they also recruit maturation and quality control factors to facilitate correct folding[40]. Consequently, mutations that disrupt glycosylation sites can disturb the folding process and cause ER retention. Our experiments show that ER retention is not a direct cause of the missing glycosylation on Asn39, as surface expression is restored in the S37A/T41I double mutation. A possible explanation for the T41I-induced ER retention could be that the folding of T41I mutant is slightly slower when Asn39 is not glycosylated. This will allow the attachment of oligosaccharides to Asn35 and since this residue is important for intra-molecular interactions the aberrant glycosylation will not allow correct folding of AMN. In our system, the T41I mutation does not result in complete knockdown of cubam surface expression likely because not all AMN are glycosylated on Asn35. However, in IGS patients carrying the T41I mutation, reduced cubam surface expression may be sufficient for causing

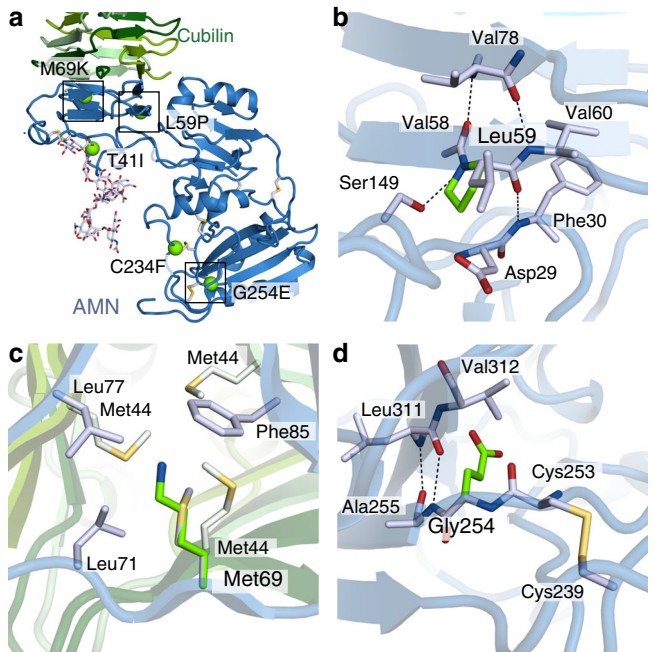

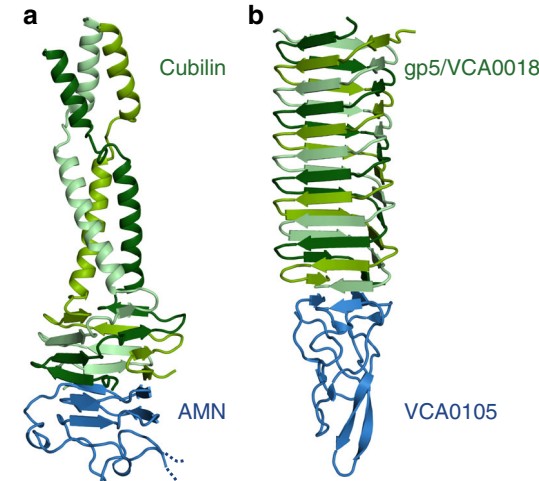

**Fig. 7** Comparison of the AMN–cubilin complex with the *V. cholerae* spike protein VCA0018/VCA0105. **a** Cartoon representation of AMN β-helix 1 (blue) in complex cubilin (residues 36–135, shades of green). **b** Cartoon representation of *V. cholerae* VCA0105 (blue) in complex with the gp5/VCA0018 fusion protein (shades of green)

**Fig. 6** Missense mutations of AMN causing Imerslund–Gräsbeck syndrome. **a** AMN missense mutations causing Imerslund–Gräsbeck syndrome are marked by green spheres on a cartoon representation of the ectoAMN–cubilin structure. **b–d** Close-up views of AMN Leu59, Met69 and Gly254. The mutations are modelled using the rotamers with least clashes and shown as green sticks. Hydrogen bonds and ionic interactions are marked by dashed black lines

the disease. Additionally, the lack of glycosylation on Asn39 could make the receptor more vulnerable to proteolytic cleavage by membrane-associated proteases on epithelial cells in the terminal ileum and therefore lead to increased shedding and further decreased surface expression of the receptor.

Our negative stain EM data shows that cubam is capable of forming dimeric structures with parallel stems. If this dimerisation occurs in vivo, cubam may be able to form a 2.8 MDa receptor complex composed of six cubilin and two AMN sub-units. Such a complex is even larger than the 600 kDa megalin, an endocytic receptor co-expressed with cubam in the kidney[41,42]. Cubam and megalin co-localise in the endocytic apparatus and appear to be functionally dependent on each other, at least in the kidney[43,44]. However, megalin is not required for cubam surface expression[16] and the fact that megalin seems less expressed in the intestine[45] suggests that cubam is capable of functioning independently of megalin. The extraordinary sizes of receptor ectodomains in both megalin and cubam are remarkable and may be advantageous for optimal catching and endocytosis of multiple ligands from the extracellular fluids.

Ca²⁺ plays an important role in many endocytic receptors and functions as a mediator of ligand binding and release[12,46–48]. On the cell surface, Ca²⁺ facilitates ligand binding, but after endocytosis Ca²⁺ is pumped out of the endosomes, which together with a lowering of the endosomal pH triggers release of ligands. The receptors are followingly recycled back to the cell surface and can now engage in another cycle of endocytosis. Although the Ca²⁺-dependent endocytic receptors are built from structurally distinct domains (e.g. CUB-domains, LA-modules and scavenger receptor cysteine-rich domains), the role of Ca²⁺ in ligand binding appears to be similar. Ca²⁺ indirectly mediates ligand binding by coordinating acidic residues from the receptor for

interaction with a positively charged residue from the ligand[12,47]. In this way Ca²⁺ functions as a "molecular lynchpin" for ligand binding/release. In cubilin, consensus Ca²⁺-binding sites are found in 13 of the 27 CUB-domains (domains 2, 5–9, 11–14, 20, 24 and 26). Several of these domains are probably binding the various ligands reabsorbed in the kidney. The characteristic crown-region of cubam as observed in EM is dependent on the presence of Ca²⁺, as complexing Ca²⁺ with EDTA causes it to fall apart. We speculate that Ca²⁺-dependent interactions also mediate interactions between the individual cubilin subunits in a similar way as ligands are bound. In the structure of CUB₅₋₈ in complex with IF the Ca²⁺ sites of CUB₆ and CUB₈ binds IF, whereas the Ca²⁺ sites of CUB₅ and CUB₇ points in the opposite direction and could very well be involved in such inter-subunit interactions.

The interdependence of AMN and cubilin has been observed across mammalian species as well as in *Drosophila melanogaster*[13,49,50], indicating that the cubam receptor architecture is of ancient origin. This is supported by sequence alignment of AMN β-helix 1 and the N-terminal region of cubilin from distantly related species (Fig. 3a, b), which shows that the hydrophobic residues forming the core of the β-helices are conserved across the species. Hence, the β-helix–β-helix association revealed in the structure of AMN–cubilin is conserved from insects to man.

## Methods

**Cloning and expression of AMN(20–357)–cubilin(26–135).** For co-expression of the ectodomain of human AMN (residues 20–357) and the N-terminal fragment of cubilin (residues 26–135), a pACYCDuet-1 vector (Novagen) was constructed using the In-Fusion HD cloning kit (Clontech). A thioredoxin–AMN construct was cloned into the EcoRI site of multiple cloning site 1 and the cubilin fragment was cloned into the NdeI site of multiple cloning site 2 (Supplementary Table 1). A tobacco etch virus (TEV) enteropeptidase site was inserted between the thioredoxin and AMN gene for proteolytic digestion of the expressed proteins. The AMN–cubilin pACYCDuet-1 vector was transformed into ShuffleT7 *E. coli* cells (New England Biolabs Inc.) for expression of native protein and B834(DE3) *E. coli* cells (Novagen) for seleno-methionine-substituted protein. Cell cultures were grown in 2xTY growth medium (for native protein) or minimal medium containing seleno-L-methionine (for seleno-methionine-substituted protein) with 34 µg ml⁻¹ chloramphenicol and induced overnight at 20 °C with 1 mM IPTG. The cells were resuspended 20 mM HEPES pH 7.6, 500 mM NaCl, 5 mM Imidazole, 10% glycerol (lysis buffer) and lysed by sonication. The lysate was centrifuged at

30,000×g for 20 min and the supernatant loaded on a 5 ml cOmplete™ His-Tag Purification Column (Roche) equilibrated in lysis buffer. The protein was eluted by running a linear gradient from 5 to 500 mM imidazole and exchanged into a buffer containing 20 mM HEPES pH 7.6, 250 mM NaCl, 10% glycerol using a PD-10 Desalting Column (GE Healthcare). TEV protease was added in a 1:30 (w/w) ratio and incubated overnight at 4 °C. The cleaved protein was loaded on 5 ml cOmplete™ His-Tag Purification Column equilibrated in 20 mM HEPES pH 7.6, 250 mM NaCl, 10% glycerol and flow-through collected. The sample was diluted 1:5 in 20 mM HEPES pH 7.6, 10% glycerol prior to loading on a 1 ml Source15Q column (GE Healthcare) equilibrated in 20 mM HEPES pH 7.6, 50 mM NaCl, 10% glycerol and flow-through collected. The sample was further purified using a 24 ml Superdex 200 column (GE Healthcare) equilibrated in 20 mM HEPES pH 7.6, 75 mM NaCl. Fractions containing pure amnionless–cubilin complex were pooled and concentrated to 5 mg ml$^{-1}$ using a Vivaspin 500 centrifugal concentrator (GE Healthcare).

**Crystallisation and data collection.** Crystals of the AMN–cubilin complex were obtained at 4 °C using the sitting-drop vapour diffusion mixing 2 μl protein solution (5 mg ml$^{-1}$) with 2 μl reservoir solution containing 1.5 M magnesium sulphate and 0.1 M MES pH 5.5. Prior to flash-freezing in liquid nitrogen, crystals were transferred to cryo-protection buffer containing 1.5 M magnesium sulphate, 0.1 M MES pH 5.5 and 25% glycerol. A native dataset extending to 2.3 Å resolution was collected at 100 K and a wavelength of 0.978 Å at the ID29 beamline at the European Synchrotron Radiation Facility (Grenoble, France). A single wave-length anomalous dispersion (SAD) dataset extending to 2.7 Å resolution from seleno-methionine derivatised crystals was collected at 100 K and wavelength 0.980 Å at the P13 EMBL beamline at Deutsches Elektronen Synchrotron (Hamburg, Germany). Diffraction data were processed and scaled using the XDS package[51]. Experimental phasing, automated model building and refinement were performed using PHENIX[52]. Manual model building was done in COOT[53] and figures were prepared with PYMOL[54].

**Purification of cubam from porcine kidneys.** Small pieces of cortex were dissected from porcine kidneys and immediately flash-frozen in liquid nitrogen. The cortex was homogenised in a buffer containing 125 mM sucrose and 0.1 mM phenylmethylsulfonyl fluoride (PMSF) (1:5 w/v) and centrifuged at 1000×g for 4 min. The supernatant was further centrifuged at 80,000×g for 40 min and after resuspension in 125 mM sucrose and 0.1 mM PMSF the pellet was rehomogenized. The homogenate was centrifuged at 80,000×g for 40 min and the pellet solubilized in 10 mM HEPES pH 7.4, 140 mM NaCl, 2 mM CaCl₂, 1 mM MgCl₂ and 1% Triton X-100. For affinity chromatography, a column containing immobilised human IF-B₁₂ was prepared. IF-B₁₂ was expressed in *Pichia pastoris* and purified using hydrophobic interaction and size-exclusion chromatography[12]. The solubilised membranes were applied on the IF-B₁₂ column equilibrated in 10 mM HEPES, 140 mM NaCl, 2 mM CaCl₂, 1 mM MgCl₂ and 0.3% CHAPS. Bound proteins were eluted with a buffer containing 10 mM HEPES, 140 mM NaCl, 2 mM CaCl₂, 1 mM MgCl₂ and 10 mM EDTA. Eluted fractions were immediately added 4:1 v/v 100 mM Tris–HCl pH 7.6, 10 mM CaCl₂. Samples containing the cubam receptor were further purified on a 24 ml Superose 6 column (GE Healthcare) equilibrated in 20 mM Tris–HCl pH 7.6, 150 mM KCl, 5 mM CaCl₂ and 0.3% CHAPS or 20 mM Tris–HCl pH 7.6, 150 mM KCl, 5 mM CaCl₂ and 10 mM EDTA.

**Negative stain EM on full-length cubam.** Fractions containing essentially pure cubam receptor were applied to freshly charged carbon-coated copper grids and allowed to adhere for 15 s before being blotted from the side. Immediately after blotting, 3 μl of 2% uranyl formate solution was applied to the grid and blotted from the side. This was repeated three times. Images were acquired on a Tecnai Spirit TEM (Thermo Fisher Scientific) equipped with a Tietz F416 charge-coupled device (CCD) camera at a nominal magnification of ×67,000 and a defocus of −1.0 μm using the Leginon data collection software[55].

**Generation of AMN mutants.** cDNA encoding AMN(1–454) was inserted in-frame with the V5-tag in the pEF4/V5–6H plasmid using EcoRI and XbaI restriction enzyme sites and was used as a template to generate the AMN mutants using the site-directed, ligase-independent mutagenesis (SLIM) method (Supplementary Table 1)[56].

**Immunoprecipitation of AMN mutants and FACS analysis.** CHO K1 cells (Thermo Fisher Scientific) were transiently transfected with cubilin cDNA encoding amino acids 1–1389 inserted into multiple cloning site of pcDNA3.1(-) using XbaI and HindIII restriction enzyme sites[13], V5-tagged AMN, or cubilin and V5-tagged AMN using Lipofectamine2000 (Thermo Scientific) and incubated 48 hours. Following, cells were either solubilized for immunoprecipitation of AMN and cubilin complexes or used for flow cytometric analysis of cubilin surface expression. For immunoprecipitation, transfected cells were solubilized with 1% Triton X-100 (Boehringer, Germany) in 1× PBS pH 7.4 supplemented with cOmplete™ mini protease inhibitor cocktail (Roche) for 5 min at 4 °C and cleared by centrifugation at 16,000×g for 10 min. Cleared lysate was incubated with 10:1 (v/v) prewashed anti-V5 agarose beads (Sigma) on an end-over-end rotator for

2 hours at room temperature. After extensive washing in PBS pH 7.4 with 0.05% triton X-100, bound material was eluted in 1x LDS loading buffer (Invitrogen) by 3 min incubation at 95 °C. Eluted protein were separated by electrophoresis on NuPAGE 4–12% Bis–Tris polyacrylamid gel and transferred to a PVDF membrane using the iBlot unit (Thermo Fischer Scientific) and followed by incubation with mouse monoclonal anti-V5 alkaline phosphatase (AP)-conjugated antibody diluted 1:5000 (Thermo Fischer Scientific) and AP substrate BCIP/NBT (Roche) for visualisation of V5-tagged proteins, or with rabbit polyclonal anti-rat cubilin antibody[44] at 1 μg ml$^{-1}$ followed by Horse-radish peroxidase-conjugated goat polyclonal anti-rabbit IgG for visualisation of cubilin. Immunoreactive bands were visualized on a FUJI FLA3000 Gel Doc (Fujifilm Europe GmbH, Düsseldorf, Germany) using ECL substrate (Pierce). Flow cytometric analysis of cubam surface expression on transfected CHO K1 cells were done by staining with Zenon Alexa647 (Molecular Probes) labelled anti-cubilin at 1 μg ml$^{-1}$ and Alexa488-labelled anti-V5 antibody (Clone 2F11F7, Thermo Fisher Scientific, 1:5000 dilution). In short, transfected cells were detached by incubation with accutase (Sigma) for 10 min at 37 °C and split into two flow tubes for separate staining of surface and intracellular proteins. Surface staining was done by incubating suspension cells at 4 °C for 1 hour with specified antibodies in 1× PBS + 2% FCS. For staining of intracellular antigens cells were first fixed and permeabilized in 4% paraformaldehyde (15 min, room temperature) and 0.1% Triton X-100 in PBS (10 min, room temperature) prior to incubation with antibodies. Stained cells were measured on a BD FACSCalibur™ flow cytometer (BD Bioscience) and data were analysed and gated using FlowJo software for Mac ver. 10.4.

**Expression and purification of AMN mutations.** The AMN T41I, L59P, M69K, C234F and G254E mutant constructs were generated with the In-Fusion HD cloning kit (Clontech) using the AMN(20–357)–cubilin(26–135) pACYCDuet-1 vector as template (Supplementary Table 1). The AMN–cubilin pACYCDuet-1 vector was transformed into ShuffleT7 *E. coli* cells (New England Biolabs Inc.). Cell cultures were grown in 2xTY growth medium with 34 μg ml$^{-1}$ chloramphenicol and induced overnight at 20 °C with 1 mM IPTG. The cells were resuspended 20 mM HEPES pH 7.6, 500 mM NaCl, 5 mM Imidazole, 10% glycerol (lysis buffer) and lysed by sonication. The lysate was centrifuged at 30,000×g for 20 min and the supernatant loaded on a 5 ml cOmplete™ His-Tag Purification Column (Roche) equilibrated in lysis buffer. The protein was eluted by running a linear gradient from 5 to 500 mM imidazole.

**Generation of AMN(20–357) and cubilin(26–135) antibodies.** Antibodies against AMN(20–357) and cubilin(26–135) were generated by immunising two mice subcutaneously two times with 14 days interval using 40 μg AMN(20–357)-cubilin(26–135) complex mixed with GERBU™ adjuvant for each injection. Two weeks later the mice received an intravenous injection of 25 μg AMN (20–357)–cubilin(26–135) complex without adjuvant. The fusion was carried out 3 days later by mixing the isolated lymphocytes from the two spleens with SP2/O-AG14 cells using Polyethylene glycol (Sigma) as fusogen. Positive hybridomas was selected by ELISA and cloned at least three times before antibody production in flasks. Two clones against AMN(20–357) and cubilin(26–135) were selected and named AMN 17-44-3 and cubilin 17-44-5, respectively.

**Analysis of mutant AMN expression.** Eluted proteins were separated by electrophoresis on NuPAGE 4–12% Bis–Tris polyacrylamid gel and transferred to a PVDF membrane using the iBlot unit (Thermo Fischer) and followed by incubation with AMN 17-44-3 (1:3000) or cubilin 17-44-5 (1:3000) followed by Horse-radish peroxidase conjugated goat polyclonal anti-mouse IgG (1:30,000). Immunoreactive bands were visualized on a FUJI FLA3000 Gel Doc (Fujifilm Europe GmbH, Düsseldorf, Germany) using ECL substrate (Pierce).

## Data availability

Atomic structure factors and coordinates have been deposited at the Protein Data Bank under accession number 6GJE. Other data are available from the corresponding author upon reasonable request. A reporting summary for this Article is available as a Supplementary Information file.

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

## Acknowledgements

We are grateful to G. Ratz for technical assistance and the staff at European Synchrotron Radiation Facility and Deutsches Elektronen Synchrotron beamlines for help with data collection. We thank T. Boesen for assistance with electron microscopy. The research was supported by Fabrikant Vilhelm Pedersen og Hustrus Mindelegat, The Novo Nordisk Foundation and The Danish Council for Independent Research.

## Author contributions

C.L.: cloning, purification, crystallisation, data collection, structure determination and analysis. A.E.: cloning, mutagenesis, flow cytometry and immunoprecipitation. M.M.: cloning, mutagenesis, immunoprecipitation and study design. K.S.: antibody generation.

S.K.M.: manuscript preparation and study design. C.B.F.A.: structure determination and analysis, electron microscopy, manuscript preparation and study design.

## Additional information

**Competing interests:** The authors declare no competing interests.

