## [Peer Review File · Nature Communications]

Reviewers' comments:

Reviewer #1 (Remarks to the Author):

In this manuscript by Larsen et al., the high-resolution crystal structure of the complex of the AMN ectodomain and N-terminal fragment of cubilin is reported as well as the negative staining of the complete cubam receptor. The structure is of high quality and together with the performed mutagenesis and flow-cytometry experiments allowed the authors to pinpoint possible roles of mutations leading to IGS syndrome. The organisation of interface between AMN and cubilin based on the beta-helices packing is really fascinating from the structural point of view. However the reported data will be interesting not only for the structural biologists, but to much wider scientific community.

Still the manuscript needs a bit more of proof-reading before the acceptance, since there are numerous typos present.

Line 45, there is unnecessary full stop after 'As. cubulin and AMN.'

Line 73 the word 'part' is missing after the 'C-terminal': 'in the C-terminal of AMN...'

Line 87 unnecessary 'in' in '...helix 1 and in engages...'

Lines 88-89 two ways of spelling" 'solvent exposed' and 'solvent-exposed'

Line 100 'b-factors' should be 'B-factors'

Lines 245, 252 'Ans39' should be 'Asn39'

Line 303: please indicate how many His residues instead of polyhisitidine

Lines 335-336 I guess you need to mention that P13 is an EMBL beamline

Lines 351, 369, 381 - three different ways of spelling Triton X-100

Line 353 'sigma' should be 'Sigma'

Lines 351, 354 'pH 7,4' should be 'pH 7.4'

Line 355 '0,05%' should be '0.05%'

Line 369 '0,1%' should be '0.1%'

Line 380 'pH 7-4' should be 'pH 7.4'

Lines 381, 383 'MgCl2' should be 'MgCl₂'

Line 388 'CaCl2' should be CaCl₂

Line 392 should be 'were' instead 'was'

Ref #50, you could use a newer recommended reference: Kabsch, W. XDS. Acta Cryst. D66, 125-132 (2010).

the same for #52 doi:10.1107/S0907444910007493

In Table 1, instead of 'a=70.3 b =158.8 c=237.2' you could just say '70.3, 158.8, 237.2'

instead of 'αβγ=90' use the standard way '90.0, 90.0, 90.0'

Which Rfactor was used Rsym or Rmerge? (i.e. keep only one)

What was the reason to truncate the data at 2.3 Å resolution? It looks as if the crystal could diffract more.

Are the structural images properly rayed? Check the quality.

Please consider in the structural figures removing NH and CO groups of main chain, which are not involved in any interactions for clarity.

In Figure 5, please use more contrast color for the mutations - red?

Reviewer #2 (Remarks to the Author):

The authors presented an elegant structural study of the interface between amnionless (AMN) and cubilin, both components of the cubam receptor involved in the intestinal uptake or kidney reabsorption of vitamin B12. Coupled with negative stain electron microscopy of the entire cubam receptor, as well as characterization of AMN variants that lead to genetic diseases, this study provided novel findings of the cubam receptor architecture, crucial for the vitamin B12 journey into the human cell.

The authors should address the following few issues for the manuscript before being considered for publication in Nat Comm.

- page 7, line 155: provide a table (e.g. supplementary data), or cite a relevant mutation review of known IGS causing mutations, as not everyone will have access to an up-to-date version of HGMD. Also state the version of HGMD that the authors interrogated.

- page 7, line 161: Briefly explain what makes T41I a predominant mutation

- pages 7-8: i have several issues with the description and presentation of the 'IGS mutations' section

(1) On several occasions the authors reported observations on glycosylations that I couldn't find the underlying experimental data . This includes statements that T41I prevents glycosylation at site II (lines 165, 177) but results in glycosylation at site I (line 251), and that N35Q and S37A prevent glycosylation at site I (lines 168, 175). This reviewer did not follow how/where these statements are derived. Is there more direct evidence that glycosylation took place in their recombinant samples, beyond the slight bandshift in SDS-PAGE Figure 4C (which, to this reviewer, is rather circumstantial)? Without concrete glycosylation demonstrated, this reviewer cannot follow the interpretation of the FACS results in Figure 4B to differentiate the behaviour among N35Q, S37A, T41I .

(2) I find the description of 'IGS mutations' section rather lengthy, with substantial scope given to speculative interpretation based on the FACS and structural inspection. The authors should consider moving some of these interpretation into discussion.

(3) Figure 5 B/C/D is quite confusing because for each mutation site (Leu59, Met69, Gly254) both wild-type and mutated side-chains are shown on top of each other, with the SAME colour scheme. This would be confusing to a structural biologist, let alone a lay researcher.

- page 10 In the discussion the authors described their structure as a completely novel architecture involving beta-helices that are not well characterized in eukaryotes. Consider bring this point out in the Results (through structural homologue analysis e.g. DALI?)

- page 32: Consider moving figure 6 earlier in the manuscript, as sequence conservation information is quite helpful when reading about the structural description

Reviewer #3 (Remarks to the Author):

In this manuscript, the authors performed structural analysis of amnionless and cubilin complex and revealed the assembly of three cubilin combined with beta-helix of amnionless. These findings were supported by the results of negative staining EM which partially overlap with the findings previously reported by Lindblom et al. in 1999.

The authors also analyzed the effects of AMN T41I and other mutations on its structure and glycosylation. These results uncover the structural basis for endocytosis in kidney and intestine. This reviewer mainly discuss the cell biology data presented in this work.

Major points:

In this manuscript, amnionless T41I mutation caused only moderate decrease in amnionless-cubilin interaction and cubilin surface expression. However, the previous study by Udagawa et al. showed that T41I mutation resulted in complete abrogation of interaction with cubilin and dramatic decrease in cubilin surface expression. To analyze the cause of these differences, the experiments using other cell lines originated from epithelial cells of kidney tubules or intestine should be performed.

The results of Figure 4, especially those of combination of mutations S37A and T41I are unexpected and interesting. In the discussion section (Page 11), the authors present several mechanistic hypotheses on this result. Further analysis will help to validate these hypotheses. For example, possibility of alternative cleavage can be examined biochemically. Are there differences in the amount of cubilin protein expression or cubilin in the cultured medium among cells expressing wild-type or each mutant amnionless? What happens when mutations at N39 (e.g. N39Q) are combined with S37A or N35W?

The result of mutant amnionless (S37A and T41I) indicated that glycosylation of amnionless is not required for cubilin surface expression. Does amnionless glycosylation have effect on cubilin glycosylation? Regarding cubilin glycosylation, the upper band of cubilin in Figure 4C which is supposed to correspond to its glycosylated form seems decreased by combined mutations of S37A and T41I, despite the unaltered membrane expression of cubilin (Figure 4B). The quantification of glycosylated cubilin as well as whole cubilin, and the intracellular localization of cubilin will elucidate the effects of combined mutation on maturation of cubilin-amnionless complex.

One question with respect to the structural analysis is whether formation of trimer of cubilin is dependent on amnionless. Is it possible to divide the role of interaction between amnionless and cubilin, and those between inter-cubilin on maturation of cubilin complex?

Minor points:

There seems to be no explanation of cubilin construct used in mammalian cell experiments (Figure 4).

As the difference in molecular weight of amnionless (Figure 4C) is hardly distinguishable, the figure should be modified also by adding the molecular weight of glycosylation-free amnionless by tunicamycin or other procedure.

In page 11 (line 242), "C243F and G354Q" should be altered to "C234F and G254F".

Reviewers' comments:

Reviewer #1 (Remarks to the Author):

In this manuscript by Larsen et al., the high-resolution crystal structure of the complex of the AMN ectodomain and N-terminal fragment of cubilin is reported as well as the negative staining of the complete cubam receptor. The structure is of high quality and together with the performed mutagenesis and flow-cytometry experiments allowed the authors to pinpoint possible roles of mutations leading to IGS syndrome. The organisation of interface between AMN and cubilin based on the beta-helices packing is really fascinating from the structural point of view. However the reported data will be interesting not only for the structural biologists, but to much wider scientific community.

Still the manuscript needs a bit more of proof-reading before the acceptance, since there are numerous typos present.

Line 45, there is unnecessary full stop after as 'As. cubulin and AMN..'

Line 73 the word 'part' is missing after the 'C-terminal': 'in the C-terminal of AMN...'

Line 87 unnecessary 'in' in '...helix 1 and in engages...'

Lines 88-89 two ways of spelling" 'solvent exposed' and 'solvent-exposed'

Line 100 'b-factors' should be 'B-factors'

Lines 245, 252 'Ans39' should be 'Asn39'

Line 303: please indicate how many His residues instead of polyhisitidine

Lines 335-336 I guess you need to mention that P13 is an EMBL beamline

Lines 351, 369, 381 - three different ways of spelling Triton X-100

Line 353 'sigma' should be 'Sigma'

Lines 351, 354 'pH 7,4' should be 'pH 7.4'

Line 355 '0,05%' should be '0.05%'

Line 369 '0,1%' should be '0.1%'

Line 380 'pH 7-4' should be 'pH 7.4'

Lines 381, 383 'MgCl2' should be 'MgCl₂'

Line 388 'CaCl2' should be CaCl₂

Line 392 should be 'were' instead 'was'

We thank the reviewer for the corrections. The typos are corrected in the revised manuscript.

Ref #50, you could use a newer recommended reference: Kabsch, W. XDS. Acta Cryst. D66, 125-132 (2010). the same for #52 doi:10.1107/S0907444910007493

The suggested references have been updated in the revised manuscript.

In Table 1, instead of 'a=70.3 b =158.8 c=237.2' you could just say '70.3, 158.8, 237.2'
instead of ' $\alpha\beta\gamma=90$ ' use the standard way '90.0, 90.0, 90.0'

Table 1 has been updated in the revised manuscript.

Which Rfactor was used R_{sym} or R_{merge}? (i.e. keep only one)

The value written in table 1 is R_{sym}. Table 1 has been updated in the revised manuscript.

What was the reason to truncate the data at 2.3 Å resolution? It looks as if the crystal could diffract more.

When processing the data to 2.1 we observe a dramatic decrease of I/sigma and increase of R_{sym} in the outer shells. Furthermore, refining the structure against this dataset does not improve the map or refinement statistics. Therefore, we have chosen to set the cutoff at 2.3.

Are the structural images properly rayed? Check the quality.

Images have been ray-traced in pymol, but at low resolution. High-resolution ray-traced images will be submitted for publication.

Please consider in the structural figures removing NH and CO groups of main chain, which are not involved in any interactions for clarity.

We acknowledge the reviewers' suggestion and have removed NH and CO groups not involved in interactions in figures 1D and 5B-D (figure 6B-D in the revised manuscript).

In Figure 5, please use more contrast color for the mutations - red?

The color scheme in figure 5 (now figure 6) has been changed in the revised manuscript.

Reviewer #2 (Remarks to the Author):

The authors presented an elegant structural study of the interface between amnionless (AMN) and cubilin, both components of the cubam receptor involved in the intestinal uptake or kidney reabsorption of vitamin B12. Coupled with negative stain electron microscopy of the entire cubam receptor, as well as characterization of AMN variants that lead to genetic diseases, this study provided novel findings of the cubam receptor architecture, crucial for the vitamin B12 journey into the human cell.

The authors should address the following few issues for the manuscript before being considered for publication in Nat Comm.

- page 7, line 155: provide a table (e.g. supplementary data), or cite a relevant mutation review of known IGS causing mutations, as not everyone will have access to an up-to-date version of HGMD. Also state the version of HGMD that the authors interrogated.

We thank the reviewer for the suggestion. A supplementary table with references has been included in the revised manuscript.

- page 7, line 161: Briefly explain what makes T41I a predominant mutation

The sentence has been rewritten.

- pages 7-8: i have several issues with the description and presentation of the 'IGS mutations' section

(1) On several occasions the authors reported observations on glycosylations that I couldn't find the underlying experimental data . This includes statements that T41I prevents glycosylation at site II (lines 165, 177) but results in glycosylation at site I (line 251), and that N35Q and S37A prevent glycosylation at site I (lines 168, 175). This reviewer did not follow how/where these statements are derived. Is there more direct evidence that glycosylation took place in their recombinant samples, beyond the slight bandshift in SDS-PAGE Figure 4C (which, to this reviewer, is rather circumstantial)? Without concrete glycosylation demonstrated, this reviewer cannot follow the interpretation of the FACS results in Figure 4B to differentiate the behaviour among N35Q, S37A, T41I .

The statements that T41I prevents glycosylation at site II and that N35Q and S37A prevents glycosylation at site I are based on disruptions of the canonical sequence motifs for N-linked glycosylations (N-x-S/T). This has been clarified in the revised manuscript and a reference has been added. In addition, evidence of glycosylation of the recombinant AMN is included by PNGaseF treatment (Figure 5C). Furthermore, the section describing the mutational and FACS data has been rewritten to clarify the issues pointed out by the reviewer.

(2) I find the description of 'IGS mutations' section rather lengthy, with substantial scope given to speculative interpretation based on the FACS and structural inspection. The authors should consider moving some of these interpretation into discussion.

We acknowledge the reviewers' suggestion. The section describing the mutations and FACS data has been rewritten and shortened in the revised manuscript. As suggested by the reviewer, the interpretation part has been moved to the discussion.

(3) Figure 5 B/C/D is quite confusing because for each mutation site (Leu59, Met69, Gly254) both wild-type and mutated side-chains are shown on top of each other, with the SAME colour scheme. This would be confusing to a structural biologist, let alone a lay researcher.

The color scheme in Figure 5 (now Figure 6) has been changed in the revised manuscript.

- page 10 In the discussion the authors described their structure as a completely novel architecture involving beta-helices that are not well characterized in eukaryotes. Consider bring this point out in the Results (through structural homologue analysis e.g. DALI?)

Running a DALI search with the beta-helical domains of AMN and cubilin did not return any significant structural homologues and is therefore not included in the manuscript. The homology to the tailspike proteins and bacterial type VI secretion system is remarkable but is not caught by DALI. The homology was discovered by browsing through the various beta-helical proteins in the PDB database. This is now described in the revised manuscript. In addition, we have added a Figure (Figure 7) showing the homologous structures.

- page 32: Consider moving figure 6 earlier in the manuscript, as sequence conservation information is quite helpful when reading about the structural description

We thank the reviewer for the suggestion and have moved the sequence conservation to Figure 3 in the revised manuscript.

Reviewer #3 (Remarks to the Author):

In this manuscript, the authors performed structural analysis of amnionless and cubilin complex and revealed the assembly of three cubilin combined with beta-helix of amnionless. These findings were supported by the results of negative staining EM which partially overlap with the findings previously reported by Lindblom et al. in 1999.

The authors also analyzed the effects of AMN T41I and other mutations on its structure and glycosylation. These results uncover the structural basis for endocytosis in kidney and intestine. This reviewer mainly discuss the cell biology data presented in this work.

Major points:

In this manuscript, amnionless T41I mutation caused only moderate decrease in amnionless-cubilin interaction and cubilin surface expression. However, the previous study by Udagawa et al. showed that T41I mutation resulted in complete abrogation of interaction with cubilin and dramatic decrease in cubilin surface expression. To analyze the cause of these differences, the experiments using other cell lines originated from epithelial cells of kidney tubules or intestine should be performed.

The study by Udagawa *et al.* investigates the effect of T41I in AMN by FACS (Figure 2f) and immunoprecipitation (Figure 5b) of transfected HEK293T cells. We agree that the immunoprecipitation experiment with cubilin-FLAG and AMN-myc-GFP fusion proteins presented in Figure 5 in Udagawa *et al.* could indicate that the AMN T41I mutation abrogates the interaction between AMN and cubilin. However, the FACS plot in Udagawa *et al.* Figure 2f shows that cells expressing AMN T41I still present cubilin at the surface albeit at a lower level. This supports our own findings in transfected CHO K1 cells. Therefore, we find that it will not add further essential information to perform expression studies in other cells lines, also in view of the structural data showing that the purified AMN T41I-cubilin expressed in *E.coli* folds correctly and forms a complex with cubilin. The latter result indicates that there are no unknown epithelial factors that control the folding of the AMN-cubilin complex.

The results of Figure 4, especially those of combination of mutations S37A and T41I are unexpected and interesting. In the discussion section (Page 11), the authors present several mechanistic hypotheses on this result. Further analysis will help to validate these hypotheses. For example, possibility of alternative cleavage can be examined biochemically. Is there differences in the amount of cubilin protein expression or cubilin in the cultured medium among cells expressing wild-type or each mutant amnionless? What happens when mutations at N39 (e.g. N39Q) are combined with S37A or N35W?

We have included PNGaseF treated AMN in Figure 4 (now Figure 5C) to show that the altered molecular weight of AMN S37A,T41I is likely due to lack of glycosylation and not a result of alternative cleavage. The hypothesis that the missing N-glycosylation on Asn39 in AMN T41I could make the receptor more susceptible to shedding (p. 11, line 260-263) is only one possibility and although interesting to pursue we believe that further biological experiments to investigate this and other mutations are outside the scope of the present study, which is primarily focused on the structural aspects of the AMN-cubilin assembly.

The result of mutant amnionless (S37A and T41I) indicated that glycosylation of amnionless is not required for cubilin surface expression. Does amnionless glycosylation have effect on cubilin glycosylation? Regarding cubilin glycosylation, the upper band of cubilin in Figure 4C which is supported to correspond to its glycosylated form seems decreased by combined mutations of S37A and T41I, despite the unaltered membrane expression of cubulin (Figure 4B). The quantification of glycosylated cubilin as well as whole cubilin, and the intracellular localization of cubilin will elucidate the effects of combined mutation on maturation of cubilin-amnionless complex.

The purpose of the present study was to generate structural information about the unique interaction between AMN and cubilin in the cubam complex and to provide potential mechanistic explanations for the IGS causing mutation that are not readily explained by other means. Although relevant to study we believe that extended focus on cubilin glycosylation is outside the scope of the present paper.

One question with respect to the structural analysis is whether formation of trimer of cubilin is dependent on amnionless. Is it possible to divide the role of interaction between amnionless and cubilin, and those between inter-cubilin on maturation of cuban complex?

So far, we have only been able to purify and crystallize the two components when they form a complex. Thus, we cannot rule out that the cubilin trimer may form in the absence of AMN. However, the fact that AMN interaction appear to affect cubilin glycosylation as shown by Udagawa *et. al.* suggests that cubilin is not properly folded on its own. This could be due to the open-ended beta-sheets as described on page 10, line 222-228 in the revised manuscript.

Minor points:

There seems to be no explanation of cubilin construct used in mammalian cell experiments (Figure4).

We apologize for the missing information. This has now been added to the material and method section p. 17 l. 374-375

As the difference in molecular weight of amnionless (Figure 4C) is hardly distinguishable, the figure should be modified also by adding the molecular weight of glycosylation-free amnionless by tunicamycin or other procedure.

We have added PNGaseF treated AMN to Figure 5C, showing that deglycosylated AMN migrates similar to AMN S37A/T41I and N35Q/T41I.

In page 11 (line242), "C243F and G354Q" should be altered to "C234F and G254F".

We have corrected the mutations in the revised manuscript.

REVIEWERS' COMMENTS:

Reviewer #2 (Remarks to the Author):

In the revised manuscript, the authors have addressed this reviewer's comments satisfactorily.